# The Premetastatic Lymph Node Niche in Gynecologic Cancer

**DOI:** 10.3390/ijms24044171

**Published:** 2023-02-20

**Authors:** Georgia Karpathiou, Fabio Orlando, Jean Marc Dumollard, Mousa Mobarki, Celine Chauleur, Michel Péoc’h

**Affiliations:** 1Pathology Department, University Hospital of Saint-Etienne, 42055 Saint-Etienne, France; 2Pathology Department, Faculty of Medicine, Jazan University, Jazan 2097, Saudi Arabia; 3Gynecology and Obstetrics Department, University Hospital of Saint-Etienne, 42023 Saint-Etienne, France

**Keywords:** myeloid-derived suppressor cells, macrophages, T cells, PD-L1, endometrial, vulvar, tenascin-C

## Abstract

It has been suggested that a primary tumor can “prepare” the draining of lymph nodes to “better accommodate” future metastatic cells, thus implying the presence of a premetastatic lymph node niche. However, this phenomenon remains unclear in gynecological cancers. The aim of this study was to evaluate lymph-node draining in gynecological cancers for premetastatic niche factors, such as myeloid-derived suppressor cells (MDSCs), immunosuppressive macrophages, cytotoxic T cells, immuno-modulatory molecules, and factors of the extracellular matrix. This is a monocentric retrospective study of patients who underwent lymph-node excision during their gynecological-cancer treatment. In all, 63 non-metastatic pelvic or inguinal lymph nodes, 25 non-metastatic para-aortic lymph nodes, 13 metastatic lymph nodes, and 21 non-cancer-associated lymph nodes (normal controls) were compared for the immunohistochemical presence of CD8 cytotoxic T cells, CD163 M2 macrophages, S100A8/A9 MDSCs, PD-L1+ immune cells, and tenascin-C, which is a matrix remodeling factor. PD-L1-positive immune cells were significantly higher in the control group, in comparison to the regional and distant cancer-draining lymph nodes. Tenascin-C was higher in metastatic lymph nodes than in both non-metastatic nodes and control lymph nodes. Vulvar cancer-draining lymph nodes showed higher PD-L1 values than endometrial cancer and cervical cancer-draining lymph nodes. Endometrial cancer-draining nodes had higher CD163 values and lower CD8 values, compared to vulvar cancer-draining nodes. Regarding regional draining nodes in low- and high-grade endometrial tumors, the former showed lower S100A8/A9 and CD163 values. Gynecological cancer-draining lymph nodes are generally immunocompetent, but vulvar cancer draining nodes, as well as high-grade endometrial cancer draining nodes, are more susceptible to harboring premetastatic niche factors.

## 1. Introduction

Knowing the status of lymph-node-draining malignancies is essential for patients’ prognosis and treatment decisions. In addition, recent studies revealed the importance of immunocompetent lymph nodes in the effectiveness of immunotherapy, since they could prevent tumor spreading [1,2,3]. Similar to “premetastatic niche” models of distant sites, according to which the primary tumor “prepares” a potential metastatic site to “better accommodate” future metastatic cells, the premetastatic lymph node niche (PLNN) model has also been suggested. The interest in characterizing a possible premetastatic lymph node niche lies in characterizing the metastatic potential of a certain tumor, since a premetastatic niche favoring the dissemination of tumor cells would constitute the first key step in the metastatic cascade, and, thus, identify the best potential molecular targets [4,5,6]. The PLNN is characterized, according to preclinical models, by the development of an immunosuppressive immune microenvironment, especially driven by myeloid-derived suppressor cells (MDSCs), which are currently promising therapeutic targets [4,5,6]. MDSCs are produced after stimulation by molecules or vesicles (exosomes) secreted by tumor cells, and they act as T-cell suppressors [6]. In addition to this immunosuppression, other factors, such as the extracellular matrix [7], are also modified in a context of the “preparation” of the tissue to receive a metastasis. In bladder cancer, tenascin-C, an extracellular matrix glycoprotein and potential therapeutic target, has been revealed to characterize the preparation of lymph nodes before the arrival of tumor cells [7].

In gynecological cancers, little data exists on PLNNs. Very recently, modified immune pathways have been identified in lymph-node draining in vulvar cancers (*n* = 25) [8]. It has also been proposed that PLNNs are responsible for false-positive PET-scan results before surgery for gynecological cancers [9]. Moreover, gynecological cancer can serve as a prototypical model for PLNNs due to the richness of the pelvic lymphatic network, often sampled routinely during the oncological surgical treatment of these tumors.

The aim of this study is to evaluate lymph-node draining in gynecological cancers, notably endometrial, cervical, and vulvar cancers, for premetastatic niche factors, such as MDSCs, immunosuppressive macrophages, cytotoxic T cells, immuno-modulatory molecules, and factors of the extracellular matrix, in order to better characterize this potentially targetable pathophysiological mechanism. These factors will be compared to non-cancer (normal) lymph nodes and metastatic lymph nodes in order to better characterize the non-metastasized cancer draining nodes.

## 2. Results

### 2.1. Cohort Studied

The patients’ characteristics are shown in Table 1. The study included 86 cancer patients and 21 non-cancer patients (control group) matched for sex and age (mean age of cancer patients: 67.725 ± 8.557, mean age of control patients: 64.333 ± 11.88, *p* = 0.1815, Fisher’s test). The cancer-patients’ cohort included 73 patients with non-metastatic lymph nodes and 13 patients with metastatic lymph nodes. Patients with non-metastatic lymph nodes included 63 patients with pelvic or inguinal lymph nodes and 25 with para-aortic nodes (for 15 patients, both lymphadenectomies were available). The primary tumors corresponded to 42 endometrial carcinomas (22 low grade, 20 high grade), 22 cervical carcinomas and 22 vulvar carcinomas. For 35 cases, all antibodies were studied bilaterally; the paired *t*-test revealed no differences between the right and left lymph nodes (*p* values for S100A8/A9, PD-L1, CD163, CD8, and tenascin-C: 0.807, 0.1105, 0.7457, 0.9241, and 0.5018, respectively); thus, one-sided, lymph-node immune-cell scores were used for further comparisons.

### 2.2. Immunohistochemical Results

The mean values of the markers studied and their comparisons according to the Fisher test are shown in Table 2, Table 3, Table 4 and Table 5. S100A8/A9 (Figure 1), CD8 (Figure 2), and CD163 (Figure 3) did not reveal statistically significant differences between the different groups studied. PD-L1-positive immune cells (Figure 4 and Figure 5) were significantly higher in the control (no cancer patients) group, in comparison to the regional and distant cancer-draining lymph nodes. Tenascin-C (Figure 6 and Figure 7) was higher in the metastatic lymph nodes than in both non-metastatic nodes and control lymph nodes.

Regarding the differences (Figure 8) in the non-metastatic regional lymph nodes between the different primaries (Fisher test), vulvar cancer-draining lymph nodes showed higher PD-L1 values than endometrial cancer-draining and cervical cancer-draining lymph nodes (9 ± 10.748 for vulva, 1.512 ± 3.333 for endometrial, and 0 for cervical, *p* = 0.0004 and 0.039, respectively). Endometrial cancer-draining nodes had higher CD163 values (6.293 ± 6.842 vs. 2.682 ± 3.920, *p* = 0.0206) and lower CD8 values (10.341 ± 7.614 vs. 25 ± 11.019 *p* < 0.0001), compared to vulvar cancer-draining nodes. Endometrial cancer-draining nodes had also lower CD8 values (10.341 ± 7.614 vs. 25.778 ± 17.042, *p* = 0.006), compared to cervical cancer-draining nodes. Tenascin-C was also higher in vulvar cancer-draining nodes (8.318 ± 10.816), compared to endometrial (2.463 ± 3.88) or cervical (0.889 ± 1.269) cancer-draining lymph nodes (*p* = 0.0007 and 0.037, respectively). For distant nodes, the only statistically significant difference regarding primaries was the CD8 values, which were higher for the cervical, compared to the endometrial primaries (23.235 ± 13.572 vs. 9.417 ± 7.621, *p* = 0.01). Regarding low- and high-grade endometrial tumors’ regional draining nodes, the former showed lower S100A8/A9 (2.273 ± 2.947 vs. 10.154 ± 6.543, *p* = 0.0001) and CD163 (5.136 ± 6.198 vs. 9.077 ± 8.46, *p* = 0.0505) values.

Lymphovascular invasion and necrosis in the primary tumor were marginally associated with higher S100A8/A9 values in the non-metastatic regional lymph nodes (6.765 ± 6.21 vs. 4.018 ± 5.176, *p* = 0.0726 and 7.5 ± 6.816 vs. 4.136 ± 5.138 *p* = 0.0549, respectively) with no other significant associations. We also sought to establish whether the lymph-node size was associated with these data: regarding the site of cancer-draining nodes, the only statistical significance was found for endometrial cancer-draining nodes which were significantly larger than the control lymph nodes (12.659 ± 5.881 vs. 9.714 ± 5.693, *p* = 0.0482) with no other differences noted. Regarding the markers studied, only the S100A8/A9 regional node values were positively associated with the node size (*p* < 0001). No association between the markers studied and the patients’ overall survival was found.

## 3. Discussion

To the best of our knowledge, this is the first study examining a large series of whole tissue sections for immunohistochemical factors associated with the premetastatic lymph node niche in gynecological cancers. We compared different primaries, left and right sides, and with non-cancer-draining nodes, elements that have not been previously studied. We showed that there is no significant difference between gynecological cancer-draining lymph nodes and control lymph nodes from non-cancer patients, which is important information when trying to answer the question of a possible premetastatic niche formation, and this question has not been previously addressed. A recent study of 25 vulvar cancer patients compared 27 non-metastatic LNs with 11 metastatic LNs [8]. In a bladder cancer patients’ study, 20 non-metastatic nodes from patients with N0 disease were compared to 20 non-metastatic nodes of otherwise N+ patients. A study on cervical cancer-draining lymph nodes compared 20 patients who had high-white-cell blood counts with 20 patients who had low-white-cell blood counts, showing higher MDSCs in the LNs of the first group [9]. An earlier study on cervical cancer compared the distant to regional cancer-draining lymph nodes [10]. In the current series, the gynecological cancer-draining LNs and the non-cancer-draining LNs differed only for the PD-L1-positive cells, which were more elevated in non-cancer draining nodes. PD-L1-positive immune cells are considered as acting as immunosuppressors for the immune microenvironment; thus, their lower numbers in cancer-draining LNs probably suggest a more immunocompetent environment. This finding, as well as the absence of any other significant difference in the factors studied, suggest that the lymph nodes that drain in gynecological cancer are generally immunocompetent, probably not acting as “efficient” premetastatic niches.

This could explain the low percentage of lymph-node metastasis in endometrial cancer: 10% for all-grade included endometrial carcinomas [11], 14% for the sentinel lymph node [12] or lymphadenectomy approaches [13], reaching almost 20% (most often micrometastases) even when introducing molecular techniques in the early-stage disease [14] and raising to 26% for high-grade cancers [15]. For vulvar cancer-draining lymph nodes, the metastatic ratio is slightly higher, 21–28% [16]. This could be in-line with the significantly higher PD-L1-positive cells in otherwise healthy vulvar cancer-draining LNs, compared to the endometrial cancer-draining LNs found here. Despite also finding lower CD163+ macrophages and higher CD8+ T cells in these nodes, compared to endometrial cancer-draining nodes which would suggest a more immunocompetent microenvironment, the presence of higher PD-L1-positive cells could denote an immunosuppressor activity. In any case, our findings suggest differences in the lymph-node premetastatic niche potential of these two primaries.

In addition, we found that low-grade endometrial cancer-draining LNs showed lower S100A/S100B MDSCs and CD163 macrophages than high-grade tumors, with no other significant differences, suggesting a more immunocompetent LN milieu in low-grade tumors. The LN size did not differ significantly between primaries or in comparison to the markers studied, suggesting that conventional imaging techniques detecting LN sizes could not predict the formation of a premetastatic LN niche. The only marker associated (positively) with LN size was S100A/S100B. In a previous study [9] of 551 gynecological-cancer (cervical, endometrial, and ovarian) patients who underwent pretreatment 18F-FDG-PET/CT (fluorine-18-fluorodeoxyglucose positron emission tomography-computed tomography) scans, a false-positive lymph-node metastasis was found in about 6–8% of patients, and this was associated with high-white-cell blood counts (TRL, tumor related leukocytosis). The authors suggested that the false-positive lymph nodes could be explained by MDSCs-mediated premetastatic niche formation using a rat model of TRL-positive and TRL-negative cervical cancer, producing false-negative lymph nodes, which were shown to have higher numbers of S100A8/A9+ cells. In addition, the authors examined S100A8/A9+ cells in the lymph nodes of 40 patients with cervical cancer showing higher counts in the TRL-positive and in the false-positive nodes. Similar to our series, where the various markers studied did not impact on prognosis, patients with true-negative nodes and patients with false-positive nodes did not differ in survival, probably indicating that the premetastatic niche alone had no negative impact, until the true metastasis is developed. In a flow-cytometry principally based study (*n* = 25) of vulvar cancer-draining LNs conducted by scrapping the cutting surface of the bisected fresh LN, more immune-suppressive features were found in the metastatic than in non-metastatic LNs studied [8]. An increased density of CD8+, FoxP3+, and PD-1+ cells was observed in sentinel LNs, compared to distant LNs of 30 patients with cervical cancer [10]. In a study of 79 endometrial cancer patients, lower LN CD169+ macrophages, a subset considered to present tumor antigens to cytotoxic lymphocytes, were found in patients with N+ disease [17].

Immune cells are not the only component suggested to participate in the premetastatic lymph node niche. In 47 patients with bladder cancer, benign perivesical nodes were examined for tenascin-C expression, an extracellular matrix glycoprotein downregulated in healthy tissues but expressed in tissue remodeling, showing that its expression was higher in LNs from patients that harbored metastasis in other LNs (N+), than in patients with no LN metastasis (N0), suggesting that this factor is a specific feature of the lymph node pre-metastatic niche [7]. This extracellular molecule has been also associated with the development of a lung premetastatic niche in an animal model of breast cancer [18]. Its expression has not been studied in the context of a gynecological-cancer premetastatic niche. In the current series, we did not observe significant increases in its expression in the different LN groups studied, suggesting that there is no extracellular matrix remodeling in these LNs. In addition to the absence of a significant immunosuppressor microenvironment, our findings probably suggest that most gynecological tumors do not provoke an important premetastatic niche in their draining nodes.

These data will add valuable information to the recently expanding field of gynecological oncoimmunology. Immunotherapy can now be used in microsatellite-unstable endometrial cancer patients regardless of the PD-L1 status and in cervical cancer patients in association with the PD-L1 status, while the important role of the immune microenvironment is constantly expanding in several gynecological conditions [19,20,21,22].

Our study suffers from certain limitations; mostly, it is a retrospective study and its largely descriptive nature does not allow for functional correlations or more detailed statistical correlations. However, it is one of the largest studies searching for these characteristics in gynecological cancer patients. We performed a comprehensive analysis of an important series of gynecological cancer-draining lymph nodes for factors associated with premetastatic niche formation, comparing with non-cancer draining nodes, the right and left sides, as well as between different primaries. We find that these lymph nodes are generally immunocompetent, but vulvar cancer-draining nodes, as well as high-grade endometrial cancer-draining nodes, are more susceptible to harboring premetastatic niche factors.

## 4. Materials and Methods

### 4.1. Study Design—Population

This was a monocentric retrospective study of patients who underwent lymph-node excision during their gynecological (endometrial, cervical, or vulvar) cancer treatment from 01/2015 to 12/2019. The inclusion criteria included the following: 1. Surgically staged carcinoma patients with either sentinel lymph-node staging (systematically treated with ultra-staging achieved by interval hematoxylin/eosin sections and cytokeratin immunohistochemistry [23]) or systematic lymph-node dissections. 2. For homogeneity reasons, only squamous cell carcinomas of the uterine cervix or the vulva were included; similarly, for endometrial cancers, in order to avoid poor reproducibility issues [24], only grade-1 endometrioid adenocarcinomas were included as representative of low-grade cancers, and only serous carcinomas and carcinosarcomas, as representative of high-grade cancers, 3. At least 2 years of follow up or until death. The local ethics committee (Terre d’Éthique, IORG0007394) approved the study (IRBN302022/CHUSTE).

The main lymph-node group (*n* = 63) studied was the non-metastatic pelvic or inguinal lymph nodes resected for sentinel lymph-node mapping or systematic lymph-node dissections (in non-successful sentinel-node bilateral mapping) for uterine or vulvar cancers, respectively. As previously reported, pelvic sentinel lymph nodes for uterine cancers vary [23]; they were in decreasing order in our study and, as defined by the surgeon: internal iliac-obturator (the distinction between them is anatomically difficult [25]), common iliac, external iliac, or labelled as pelvic without further definition. In cases of dissection, we chose the iliac-obturator section for homogeneity reasons, as this was the most frequent anatomical site.

Since the lymphatic drainage of the uterus is bilateral, both right and left sentinel lymph nodes or pelvic lymphadenectomies are performed. Given that it would be impossible to predict which site could serve as a possible premetastatic niche, we included both left and right lymph nodes in a subset of cases (*n* = 35) to compare them.

Furthermore, a group (*n* = 25) which consisted of the most distant lymph nodes (supramesenteric para-aortic nodes) draining in uterine tumors were included.

Moreover, we sought to compare these non-metastatic lymph nodes excised in gynecological cancer patients with non-cancer-associated lymph nodes (normal controls, *n* = 21). These normal lymph nodes were excised as fortuitous findings during vascular surgeries (most often internal mammary or carotid lymph nodes) and clinical files were verified to exclude patients with a previous cancer diagnosis, or suspicion of cancer or a lymphoma diagnosis. Similarly, a subset of gynecological-cancer metastatic lymph nodes (*n* = 13) was included for the purpose of comparison.

### 4.2. Immunohistochemistry

Immunohistochemistry was performed to evaluate the immune-cell populations using the following markers: CD8 for cytotoxic T cells, CD163 for M2 macrophages, S100A8/A9 complex for MDSCs [9], PD-L1 expression by lymphocytes and macrophages as immunosuppressor molecules, and tenascin-C, as a matrix remodeling factor [7]. Whole-tissue sections were studied for CD8 (C8/144B, Dako Agilent, 1/100), CD163 (10D6, Novocastra, 1/200), PD-L1 (22C3, Dako, Agilent, 1/40), S100A8/A9 (MAC387, abcam, 1/10000), and tenascin-C (EPR4219, abcam, 1/1000) using an automated staining system (OMNIS, Dako-Agilent, Santa Clara, CA, USA) and the EnVision FLEX kit (OMNIS, Dako, Denmark), according to the manufacturer’s protocol.

Immunohistochemical evaluation of each immune-cell marker was recorded as a continuous variable in a semi-quantitative manner evaluating the percentage of the lymph-node area occupied by the immune cells [26]. The intensity of the staining in these immune cells was not taken into account [26]. The evaluation was performed by three pathologists until final agreement; the whole slide was studied with a full assessment of the lymph-node surface.

### 4.3. Statistical Analysis

We used the Fisher test to compare the immune-cell values between the different lymph-node groups. For the left and right lymph-node comparisons, the paired *t*-test was used. For all analyses, the statistical significance was indicated at a *p* value of <0.05. Parametric test was preferred to avoid non-normal distribution and spread issues. As a general rule of thumb, the subjects should be at least 50 + 8xpredictors; thus, in our study, 50 + 8 × 5 = 90 cases were analyzed [27]. Data were analyzed using the StatView© software (version 5, Abacus Concepts, Berkley, CA, USA).

## Figures and Tables

**Figure 1 ijms-24-04171-f001:**
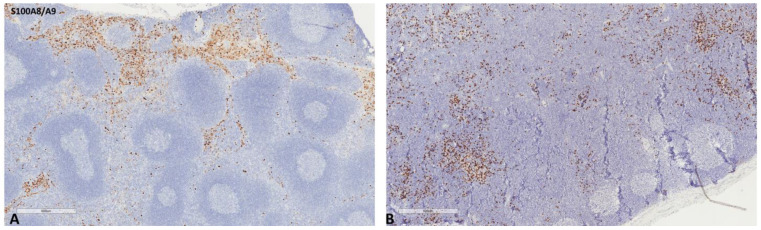
Representative microscopic images of S100A8/A9 expression in a normal cancer-draining lymph node (**A**) and a normal non-cancer-draining lymph node (**B**).

**Figure 2 ijms-24-04171-f002:**
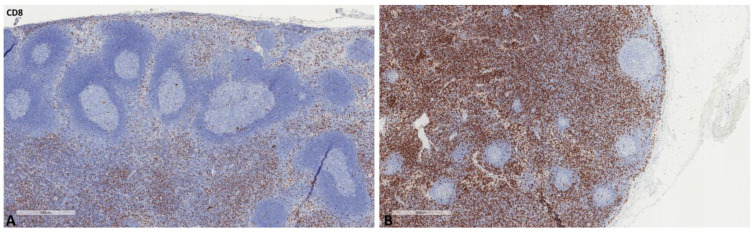
Representative microscopic images of CD8 expression in a normal cancer-draining lymph node (**A**) and a normal non-cancer-draining lymph node (**B**).

**Figure 3 ijms-24-04171-f003:**
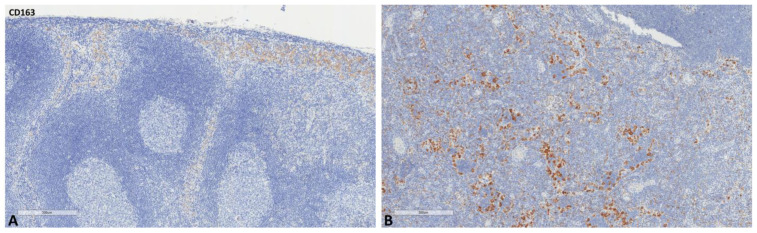
Representative microscopic images of CD163 expression in a normal cancer-draining lymph node (**A**) and a normal non-cancer-draining lymph node (**B**).

**Figure 4 ijms-24-04171-f004:**
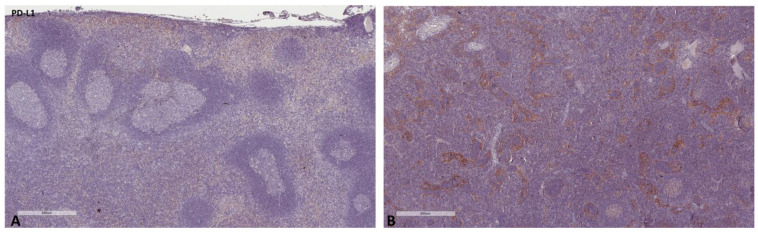
Representative microscopic images of PD-L1 expression in a normal cancer-draining lymph node (**A**) and a normal non-cancer-draining lymph node (**B**).

**Figure 5 ijms-24-04171-f005:**
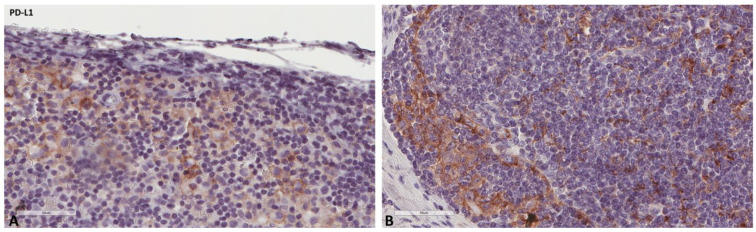
Representative microscopic images of PD-L1 expression (higher magnification) in a normal cancer-draining lymph node (**A**) and a normal non-cancer-draining lymph node (**B**).

**Figure 6 ijms-24-04171-f006:**
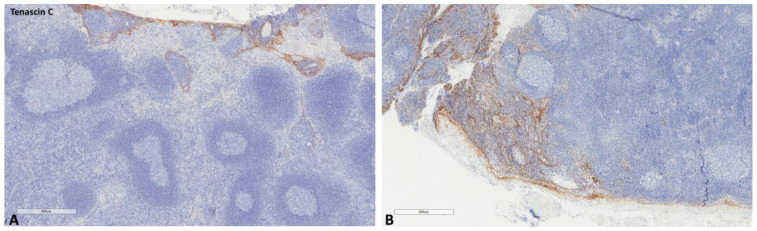
Representative microscopic images of tenascin-C expression in a normal cancer-draining lymph node (**A**) and a normal non-cancer-draining lymph node (**B**).

**Figure 7 ijms-24-04171-f007:**
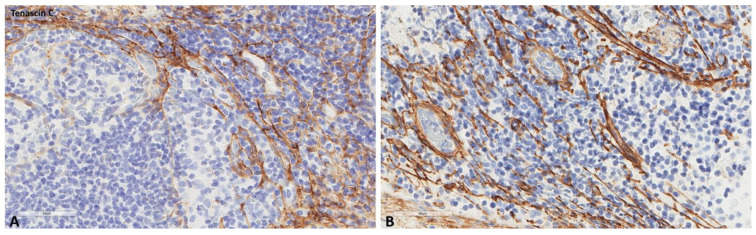
Representative microscopic images of tenascin-C expression (higher magnification) in a normal cancer-draining lymph node (**A**) and a normal non-cancer-draining lymph node (**B**).

**Figure 8 ijms-24-04171-f008:**
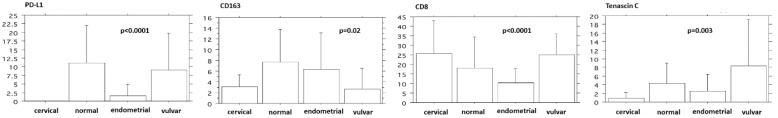
Differences in PD-L1, CD163, CD8, and tenascin-C between the various groups.

**Table 1 ijms-24-04171-t001:** Cohort’s characteristics.

Parameter	
Cohort studied (*n* = 107)	
Cancer patients	86 (80.4%)
Non-cancer patients	21 (19.6%)
Age	
Range	34–91
Mean ± SD	65.68 ± 11.77
Cancer patients (*n* = 86)	
Non-metastatic lymph nodes	73 (84.9%)
Metastatic lymph nodes	13 (15.1%)
Primary-tumor localization (*n* = 86)	
Endometrium	42 (48.8%)
Vulva	22 (25.6%)
Uterine cervix	22 (25.6%)
Primary-tumor histologic type (*n* = 86)	
Low-grade endometrial adenocarcinoma (grade 1 endometrioid adenocarcinoma)	22 (25.6%)
High-grade endometrial carcinoma	20 (23.2%)
Squamous cell carcinoma of the cervix	22 (25.6%)
Squamous cell carcinoma of the vulva	22 (25.6%)
Primary-tumor necrosis (*n* = 86)	
Yes	13 (15.1%)
No	73 (84.9%)
Primary-tumor lymphovascular invasion (*n* = 86)	
Yes	18 (21%)
No	68 (79%)
Lymph-node size (range, mean ± SD, mm)	
Regional	3–37, 11.53 ± 5.52
Distant	4–21, 10.46 ± 5.22
Follow up (months)	
Range	12–72
Mean ± SD	41.27 ± 19.37
Disease stage	
I	56, 65.1%
II	11, 12.8%
III	17, 19.8%
IV	2, 2.3%
Patients’ status	
Alive	75, 87.2%
Dead	11, 12.8%
Treatment	
Surgical treatment	30, 34.9%
Surgical and adjuvant treatments	56, 65.1%

**Table 2 ijms-24-04171-t002:** Comparison between regional cancer-draining lymph nodes and non-cancer-draining lymph nodes.

	Non-Metastatic Regional Lymph Nodes (*n* = 63)	Control Lymph Nodes (*n* = 21)	*p*
S100A/S100B	4.619 ± 5.428	4.857 ± 7.683	0.8771
PD-L1	2.921 ± 5.796	11.190 ± 10.902	0.0001
CD163	4.889 ± 5.778	7.619 ±6.136	0.0725
CD8	15.651 ± 11.83	18.095 ± 16.239	0.4656
Tenascin-C	2.429 ± 3.901	4.333 ± 4.757	0.1775

**Table 3 ijms-24-04171-t003:** Comparison between distant cancer-draining lymph nodes and non-cancer-draining lymph nodes.

	Distant Lymph Nodes (*n* = 25)	Control Lymph Nodes (*n* = 21)	*p*
S100A/S100B	3.708 ± 5.96	4.857 ± 7.683	0.5687
PD-L1	1.44 ± 3.267	11.190 ± 10.902	<0.0001
CD163	6.84 ± 6.27	7.619 ±6.136	0.6646
CD8	17.72 ± 14.073	18.095 ± 16.239	0.9318
Tenascin-C	1.84 ± 2.095	4.333 ± 4.757	0.0222

**Table 4 ijms-24-04171-t004:** Comparison between regional cancer-draining lymph nodes and metastatic lymph nodes.

	Non-Metastatic Regional Lymph Nodes (*n* = 63)	Metastatic Lymph Nodes (*n* = 13)	*p*
S100A/S100B	4.619 ± 5.428	5 ± 6.461	0.8611
PD-L1	2.921 ± 5.796	8.444 ± 13.621	0.0597
CD163	4.889 ± 5.778	4.111 ± 6.864	0.7152
CD8	15.651 ± 11.83	24.444 ± 15.092	0.0656
Tenascin-C	2.429 ± 3.901	15.444 ± 13.173	<0.0001

**Table 5 ijms-24-04171-t005:** Comparison between non-cancer-draining lymph nodes and metastatic lymph nodes.

	Control Lymph Nodes (*n* = 21)	Metastatic Lymph Nodes (*n* = 13)	*p*
S100A/S100B	4.857 ± 7.683	5 ± 6.461	0.9532
PD-L1	11.190 ± 10.902	8.444 ± 13.621	0.3986
CD163	7.619 ± 6.136	4.111 ± 6.864	0.1433
CD8	18.095 ± 16.239	24.444 ± 15.092	0.2318
Tenascin-C	4.333 ± 4.757	15.444 ± 13.173	<0.0001

## Data Availability

Data are available upon reasonable request.

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
