# Peer review of "The Premetastatic Lymph Node Niche in Gynecologic Cancer"

_ijms, 2023, doi:10.3390/ijms24044171_

Round 1

Reviewer 1 Report

The manuscript “ The premetastatic lymph node niche in gynecologic cancer” by Georgia Karpathiou and co-authors to evaluate lymph nodes draining gynecological cancers for premetastatic niche factors, such as Myeloid-Derived Suppressor Cells (MDSCs), immunosuppressive macrophages, cytotoxic T cells, immuno-modulatory molecules, and factors of the extracellular matrix. PD-L1 positive immune cells were significantly higher in the control group in comparison to regional and distant cancer-draining lymph nodes. Tenascin-C was higher in metastatic lymph nodes than in both non-metastatic nodes and control lymph nodes. Vulvar cancer draining lymph nodes showed higher PD-L1 values than endometrial cancer and cervical cancer draining lymph nodes. Endometrial cancer draining nodes had higher CD163 values and lower CD8 values compared to vulvar cancer draining nodes. Regarding low- and high-grade endometrial tumors regional draining nodes, the former showed lower S100A8/A9 and CD163 values. Gynecologic cancer draining lymph nodes are generally immunocompetent, but vulvar cancer-draining nodes, as well as high-grade endometrial cancer-draining nodes are more susceptible to harbor premetastatic niche factors. However, there are concerns which must be taken into account before the work can be reconsidered for publication.

1.      Figures1-7: Please provide more clearer photos. The negative and positive controls of immunohistochemistry staining should be provided.

2. Please describe how to evaluate each marker of immunostaining score (intensity x percentage of positive cells) in both tumor cells and immune cells.

Author Response

R1

The manuscript “ The premetastatic lymph node niche in gynecologic cancer” by Georgia Karpathiou and co-authors to evaluate lymph nodes draining gynecological cancers for premetastatic niche factors, such as Myeloid-Derived Suppressor Cells (MDSCs), immunosuppressive macrophages, cytotoxic T cells, immuno-modulatory molecules, and factors of the extracellular matrix. PD-L1 positive immune cells were significantly higher in the control group in comparison to regional and distant cancer-draining lymph nodes. Tenascin-C was higher in metastatic lymph nodes than in both non-metastatic nodes and control lymph nodes. Vulvar cancer draining lymph nodes showed higher PD-L1 values than endometrial cancer and cervical cancer draining lymph nodes. Endometrial cancer draining nodes had higher CD163 values and lower CD8 values compared to vulvar cancer draining nodes. Regarding low- and high-grade endometrial tumors regional draining nodes, the former showed lower S100A8/A9 and CD163 values. Gynecologic cancer draining lymph nodes are generally immunocompetent, but vulvar cancer-draining nodes, as well as high-grade endometrial cancer-draining nodes are more susceptible to harbor premetastatic niche factors. However, there are concerns which must be taken into account before the work can be reconsidered for publication.

 Answer: We thank the reviewer for taking the time to read our work and provide us with constructing comments.

  1. Figures1-7: Please provide more clearer photos. The negative and positive controls of immunohistochemistry staining should be provided.

 Answer: We will submit the photos as separate files, hoping that the editorial office can insert them in the text without losing in quality. We are of course at the disposition of the editors if any other files are needed. Regarding controls, the normal lymph nodes served as controls. Also, the normal inflammatory cell population of each lymph node can be used as negative or positive control for these antibodies.

  1. Please describe how to evaluate each marker of immunostaining score (intensity x percentage of positive cells) in both tumor cells and immune cells.

 Answer: Since the cells evaluated were only inflammatory cells, no intensity is counted for them, PD-L1 for example is considered positive in immune cells, regardless the intensity. This is why we gave only the perecentage of positive cells.

Reviewer 2 Report

Comments:

1. Could the authors please clarify how the numbers of study participants were determined i.e. how was their study powered?

2. Could the authors please justify the use of parametric statistical analyses please?  

Author Response

 Answer: We thank the reviewer for taking the time to read our work and provide us with constructing comments.

  1. Could the authors please clarify how the numbers of study participants were determined i.e. how was their study powered?

 Answer: Indeed, it is not easy to to predict the power of the study for retrospective studies. As a general rule of thumb (Green), 50+8×predictors subjects would suffice. However, our study still is largely descriptive, and this is now added to the limitations.

  1. Could the authors please justify the use of parametric statistical analyses please?  

 Answer: We preferred a parametric test to avoid nonnormal distribution data and different group spread issues.

Reviewer 3 Report

This is an interesting and original paper also because immuno-oncology is improving in recent years for gynecologic oncology. I think we need more research in this field to better understand which patients can benefit from immunotherapy and also the immunological mechanism involved in tumor progression. The reference list can be improved and I would suggest to add some references about gyn onc and immunological factors ( for example :

Gasparri M L et al Asian Pac J Cancer Prev 2015, Tumor infiltrating lymphocytes in ovarian cancer; others)

Author Response

This is an interesting and original paper also because immuno-oncology is improving in recent years for gynecologic oncology. I think we need more research in this field to better understand which patients can benefit from immunotherapy and also the immunological mechanism involved in tumor progression. The reference list can be improved and I would suggest to add some references about gyn onc and immunological factors ( for example :

Gasparri M L et al Asian Pac J Cancer Prev 2015, Tumor infiltrating lymphocytes in ovarian cancer; others)

 Answer:   We thank the reviewer for taking the time to read our work and provide us with constructing comments. We added a section in the discussion.

Reviewer 4 Report

The theme of the study is very intriguing, however there are some minor comments.

1. Could you explain where you obtained the lymph nodes from the non-cancer and cancer- patients? (ex. external iliac ....)

2. Could The subgroups of the cancer pts are heterogenous. It is suggested to write the histology types according to the origin of the cancer in Table 1 (Ex: cervical cancer-scc (5), adenoca (8)).

3. The authors investigated the cancer lymph node drainage of the left or right. Could specify the distance from the cancer mass to the lymph node? (or at least the location) I think the distance from the cancer would be more important than the sides.

4. Figure8 is too small please mend the figure size or quality

Author Response

The theme of the study is very intriguing, however there are some minor comments.

 Answer: We thank the reviewer for taking the time to read our work and provide us with constructing comments.

  1. Could you explain where you obtained the lymph nodes from the non-cancer and cancer- patients? (ex. external iliac ....)

 Answer: We now added the lymph nodes anatomic location.

  1. Could The subgroups of the cancer pts are heterogenous. It is suggested to write the histology types according to the origin of the cancer in Table 1 (Ex: cervical cancer-scc (5), adenoca (8)).

 Answer: This is now added.

  1. The authors investigated the cancer lymph node drainage of the left or right. Could specify the distance from the cancer mass to the lymph node? (or at least the location) I think the distance from the cancer would be more important than the sides.

 Answer: Unfortunately, the distance is not routinely recorded in these patients. However, we now added the anatomical location, that could provide an idea of the distance to the readers.

  1. Figure8 is too small please mend the figure size or quality

 Answer: we submitted separately all figures hoping that the editorial office could insert them without changing the quality.

Round 2

Reviewer 1 Report

The revised manuscript “The premetastatic lymph node niche in gynecologic cancerhave adequately addressed my previous concerns and the paper is now acceptable for publication.

Reviewer 2 Report

The authors have satisfactorily answered my comments.